# Isolated detection of elastic waves driven by the momentum of light

Tomaž Požar [1], Jernej Laloš[1], Aleš Babnik[1], Rok Petkovšek[1], Max Bethune-Waddell[2], Kenneth J. Chau[2], Gustavo V.B. Lukasievicz [3] & Nelson G.C. Astrath [4]

Electromagnetic momentum carried by light is observable through the mechanical effects radiation pressure exerts on illuminated objects. Momentum conversion from electromagnetic fields to elastic waves within a solid object proceeds through a string of electrodynamic and elastodynamic phenomena, collectively bound by momentum and energy continuity. The details of this conversion predicted by theory have yet to be validated by experiments, as it is difficult to distinguish displacements driven by momentum from those driven by heating due to light absorption. Here, we have measured temporal variations of the surface displacements induced by laser pulses reflected from a solid dielectric mirror. Ab initio modelling of momentum flow describes the transfer of momentum from the electromagnetic field to the dielectric mirror, with subsequent creation/propagation of multi-component elastic waves. Complete consistency between predictions and absolute measurements of surface displacements offers compelling evidence of elastic transients driven predominantly by the momentum of light.

[1] Faculty of Mechanical Engineering, University of Ljubljana, Ljubljana 1000, Slovenia. [2] School of Engineering, The University of British Columbia, Kelowna, BC V1V-1V7, Canada. [3] Department of Physics, Universidade Tecnológica Federal do Paraná, Medianeira, Paraná 85884-000, Brazil. [4] Department of Physics, Universidade Estadual de Maringá, Maringá, Paraná 87020-900, Brazil. Correspondence and requests for materials should be addressed to T.P. (email: tomaz.pozar@fs.uni-lj.si) or to N.G.C.A. (email: ngcastrath@uem.br)

Over a century ago, it was predicted by Maxwell in 1873[1] and Bartoli in 1876[2] that radiation pressure exists due to momentum residing in electromagnetic fields. The simplest illustration of radiation pressure is the classic case of a short plane-wave light pulse normally incident onto a flat, perfectly reflective, motionless slab. The pulse carries finite momentum in its electromagnetic fields and, upon reflection from the slab, must impart a normal mechanical momentum that is twice the incident pulse momentum to conserve total momentum. In the ideal case of a rigid slab, the normal mechanical momentum is instantaneously distributed throughout the slab and the slab moves uniformly at constant velocity along the incident pulse trajectory. In the realistic case of a deformable slab, the normal mechanical momentum is transported throughout the slab by elastic waves travelling at finite velocities. If the pulse has a finite cross-section, the pulse edges also induce elastic waves that transport a shear mechanical momentum component perpendicular to the incident pulse trajectory. Momentum-carrying elastic waves are the fundamental mechanism by which radiation pressure manifests in deformable media and play an important role in applications such as optical tweezers in soft matter[3–9].

Radiation-pressure-driven elastic waves have been recently detected in solids[10]. The physical origin of these waves was validated by agreement of measured amplitudes with a simple one-dimensional model and consistency of propagation velocity with time-of-flight predictions. In addition to amplitude and velocity, the shape of radiation-pressure-driven elastic waves gives rich insights into light–matter interaction, which could potentially be used to differentiate various theoretical formulations describing light-matter coupling. To fully understand elastic waveform features, the physics underlying their creation and propagation must be modelled and then compared to a measurable quantity, such as the material displacement induced by these transient waves.

Displacements associated with these elastic waves propagate at acoustic velocities and have peak amplitudes of the order of picometers. Experimental measurement of elastic wave propagation on absolute scales requires fast (>1 MHz bandwidth), sensitive (~10 fm resolution), and well-calibrated surface displacement detection. Even with adequate temporal and spatial resolution, isolation of momentum-driven elastic waves is difficult due to the presence of absorption-driven thermoelastic waves[11], which are typically the dominant elastic wave component created during light–matter interaction. Therefore, definitive measurement of momentum-driven elastic waves requires absorption mitigation to reduce thermoelastic wave contributions and a means of tracing different elastic wave components to either absorption or momentum transfer.

Thermoelastic wave components and momentum-driven elastic wave components can be modelled from first principles. The latter is complicated by the existence of multiple electrodynamic theories that describe the spatial distribution of normal and shear force densities in differing ways[12–26], an ongoing debate related to the Abraham–Minkowski controversy[27,28]. The most-used electrodynamic theories are the Abraham, Minkowski, Einstein-Laub, Chu, and Amperian formulations[21,22]. Using these theories in conjunction with elastodynamic theory, it is possible to simulate the shape, amplitude, and speed of momentum-driven elastic waves using the elastic properties of the medium and the properties of the incident light (i.e., without fitting parameters). Comparison between absolute surface displacement measurements and simulated displacements based on first principles provides a highly rigorous, yet unexplored, method to correlate the elastic waves in an illuminated object to the electromagnetic momentum delivered by incident light.

In this paper, we have measured elastic waves driven predominantly by the momentum of light. We use an experimental configuration in which a solid-state dielectric mirror is illuminated by short laser pulses from air, generating transient elastic waves detected as ripples on the surface of the mirror. The use of a solid-state medium enables the design of a near-perfect mirror with minimal absorption at the frequency of the incident pulse. The elastic wave amplitudes generated on solids are smaller than those previously measured in liquids[29–31], but are easier to interpret due to negligible redistribution of matter and insignificant effects of the gravitational and surface tension forces[32]. Elastic waves are captured using a piezoelectric detection method that measures surface displacement at various positions adjacent to the site of pulse reflection. Displacements have been measured with sufficient spatial (1 mm lateral and 40 fm vertical) and temporal (0.2 μs) resolution to visualize elastic wave propagation. To determine the origin of the measured waveforms, we use ab initio modelling of momentum deposition[20,21] and material deformation[11,33]. Predicted waveforms match the shape and amplitude of those measured in the experiments. The amplitude of the measured waveforms can be primarily attributed to radiation pressure, since the thermoelastic contribution to the displacements is within the uncertainty of the whole measurement chain. This work provides a method to quantitatively measure momentum coupling between electromagnetic fields and matter, which can be applied to characterize materials and to further advance optical tweezer technology[6–9]. Because elastic waves are sensitive to the distribution of normal and shear force densities as opposed to just the total force, this work has a potential to provide a means to empirically validate differing electrodynamic theories.

## Results

**Detection of optically driven elastic waves.** We observe momentum-driven elastic waves using an experimental configuration in which a pulsed axial-symmetric, linear-polarized, normal-incident, collimated light beam, propagating through air, reflects off of a planar multiple-layer dielectric mirror. This configuration maximizes the transfer of momentum along the pulse trajectory, as the mirror receives a normal component of mechanical momentum that is twice the electromagnetic momentum of the incident pulse. Due to electrostriction effects originating at the tapered part of the laser beam, the mirror is also subjected to radial electromagnetic forces, two orders of magnitude smaller than the normal force component. Mechanical momentum in the mirror manifests as elastic waves propagating at acoustic velocities. The normal component of momentum can be observed by measuring minute normal displacements of the surface adjacent to the site of reflection. To maximize the surface displacements due to momentum transfer, we tailor three aspects of the experiment: the generation of a sufficiently energetic incident pulse that is below the damage threshold of the mirror; the manufacture of a nearly-perfect mirror at the wavelength of the pulse to mitigate absorption-driven thermoelastic waves; and the optimization of a sufficiently sensitive, large-bandwidth, calibrated detector to measure dynamic surface displacements on absolute scales.

The incident pulse wavelength of 1064 nm is chosen due to availability of Q-switched, linearly polarized, high-energy laser pulses with well-behaved temporal and spatial profiles. The excitation laser pulse has an energy of 160 mJ, a circularly symmetric tapered top-hat spatial distribution with a radius of 2.05 mm, and a temporal profile in the shape of a roughly asymmetric bell with the full width at half maximum (FWHM) of 19 ns (see Methods). The pulse is delivered to the mirror through

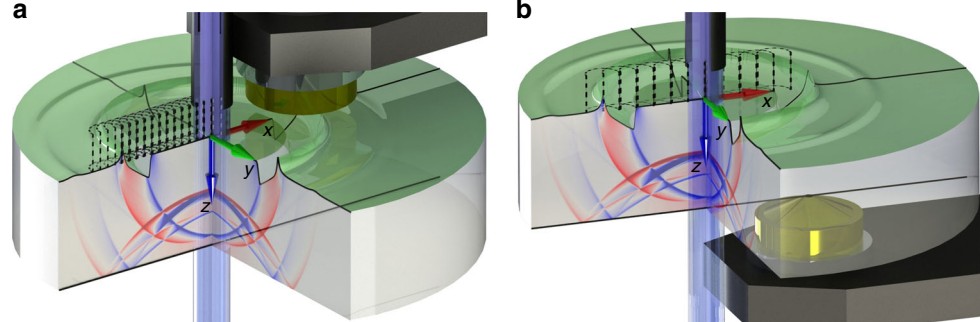

**Fig. 1** Experimental setup and geometry. The perspective view on the central part of the experimental setup with the sensor deployed on the HR surface (**a**) and on the opposite (uncoated) facet (**b**) for a beam-center to sensor-center separation of $x = 12$ mm. The $x$, $y$, and $z$ axes of the Cartesian coordinate system have the origin at the intersection of the HR surface and the symmetry axis of the beam. For visualization purposes, a part of the sample has been cutout. The laser beam pulse (indigo) impinges on the HR surface (green top layer) of the cylindrical sample (grey) through a tube (black), which prevents scattered light from reaching the sensor head (gold) fixed in the holder arm (dark grey). A minute amount of light (pale blue) is transmitted through the HR layer. The radiation pressure launches elastic waves, which spread away from the source and carry the momentum and energy transferred from the laser pulse. The calculated out-of-plane displacement at time $t = 1.96$ µs after initial pulse reflection is presented as ripples on the HR surface. The instantaneous $z$-velocity field of the bulk waves is colour-coded on the cut-out sides of the sample, where the red colour represents positive and blue negative $z$-velocity. The other experimental positions of the incident beam are depicted with the dashed contours

a tube to block Rayleigh scattered light in the surrounding air ambient. The energy of the pulse has an upper bound set by the laser induced damage threshold (LIDT) of the mirror, which in turn sets an upper bound on the displacement amplitudes of momentum-driven elastic waves.

The mirror is a dielectric high-reflective (HR) Bragg mirror with a measured normal-incidence reflectance of 99.93% at the wavelength of 1064 nm. The HR-coating consists of thin, alternating low-loss and nonmagnetic layers of $ZrO_2$ and $SiO_2$ (complete description is provided in the Methods section). Momentum transfer from the pulse occurs within the first few layers of the HR coating where the pulse is nearly totally reflected. The LIDT for the HR coating is 10 J cm$^{-2}$, which is well above the fluence of the incident pulse. The HR mirror is coated on the top face of a cylindrical fused silica substrate with a radius of 25.4 mm and a thickness of 12 mm. The low coefficient of linear thermal expansion $(5.6 \times 10^{-7}$ K$^{-1})$ of the silica substrate additionally suppresses thermally excited elastic waves.

The detector is a piezoelectric sensor with a sufficiently flat and wide frequency response to resolve normal surface displacement waveforms with amplitudes in the 1-pm range and propagating at acoustic velocities. Sensor calibration[34,35] reveals a reasonably flat frequency response between 10 kHz and 5 MHz. The circular contact tip of the conically shaped lead zirconate titanate (PZT) crystal has a 1-mm diameter and is covered by a protective nickel foil. As its aperture is not infinitesimally small, the detected displacements are spatially averaged into a single output signal. Due to the relatively low reflectivity of nickel ($R = 73\%$ at 1064 nm), a soft gold foil ($R = 99\%$ at 1064 nm, thickness: 20 µm) is sandwiched between the sensor tip and the substrate serving both to further suppress absorption of the laser pulse at the sensor tip and to provide a good mechanical contact. Additionally, the head of the sensor is coated with gold to increase the reflectance of stray light (see Methods).

The incident light pulse is directed at normal incidence onto the HR-coated surface. Surface displacements are measured on two surfaces: the illuminated HR-coated surface and the bottom (uncoated) face of the substrate. Surface displacements on the HR-coated surface are measured at 15 positions where the lateral distance from the centre of the illuminated spot ranges from 12 to 26 mm in 1-mm increments. Surface displacements on the opposite (uncoated) side of the substrate are measured at 11

positions where the lateral distance ranges from 4 to 24 mm in 2-mm increments. The experimental setup is illustrated in Fig. 1.

At each sensor position, dynamic surface displacements are recorded 200 times with a temporal resolution of 2 ns and a duration of 9 µs. The recording duration allows for elastic transients to reverberate and traverse between the top and bottom surfaces of the cylindrical substrate up to four times. Recordings are concluded before the Rayleigh wave reflected from the curved sides of the substrate reaches the sensor. Statistical analysis of the recordings establishes with nearly 100% confidence that the noise in the measurements is stochastic (see Methods). The processed sensor responses, which directly correspond to normal displacements on the illuminated substrate, are presented in two cascade plots in Fig. 2. The standard deviation in the plots corresponds to the remnant stochastic noise after averaging. The minimum detectable displacement above the noise level is 40 fm, which was achieved after averaging over 200 repeatable pulse reflections and over the aperture of the sensor.

**Simulations of momentum transfer and elastic wave creation.** We use ab initio simulations based on first physical principles to fully model the experiment. This includes the transfer of energy and momentum from the light pulse to the substrate, the creation and propagation of elastic waves that cause surface displacements on the substrate, and the resulting piezoelectric sensor signal due to these displacements. Simulated sensor signals are thus obtained without fitting parameters, but rather through a virtual experiment based on parameters that mimic the experimental configuration with high fidelity.

From the laser illumination distribution, the optical force density distributed within the HR coating is simulated by the finite-difference time-domain (FDTD) method in two dimensions (2D)[21]. This method solves the electromagnetic-material momentum continuity equation in the experimental configuration of a laser pulse incident onto a dielectric mirror consisting of alternating $SiO_2$ and $ZrO_2$ layers. For linearly polarized light, 2D force density profiles for transverse-electric (TE) and transverse-magnetic (TM) polarizations are calculated and interpolated into a full 3D solution (see Methods for more discussion on the FDTD method). The results are shown in Fig. 3.

Five optical force distributions are calculated using the following electrodynamic formalisms (see Methods): Abraham

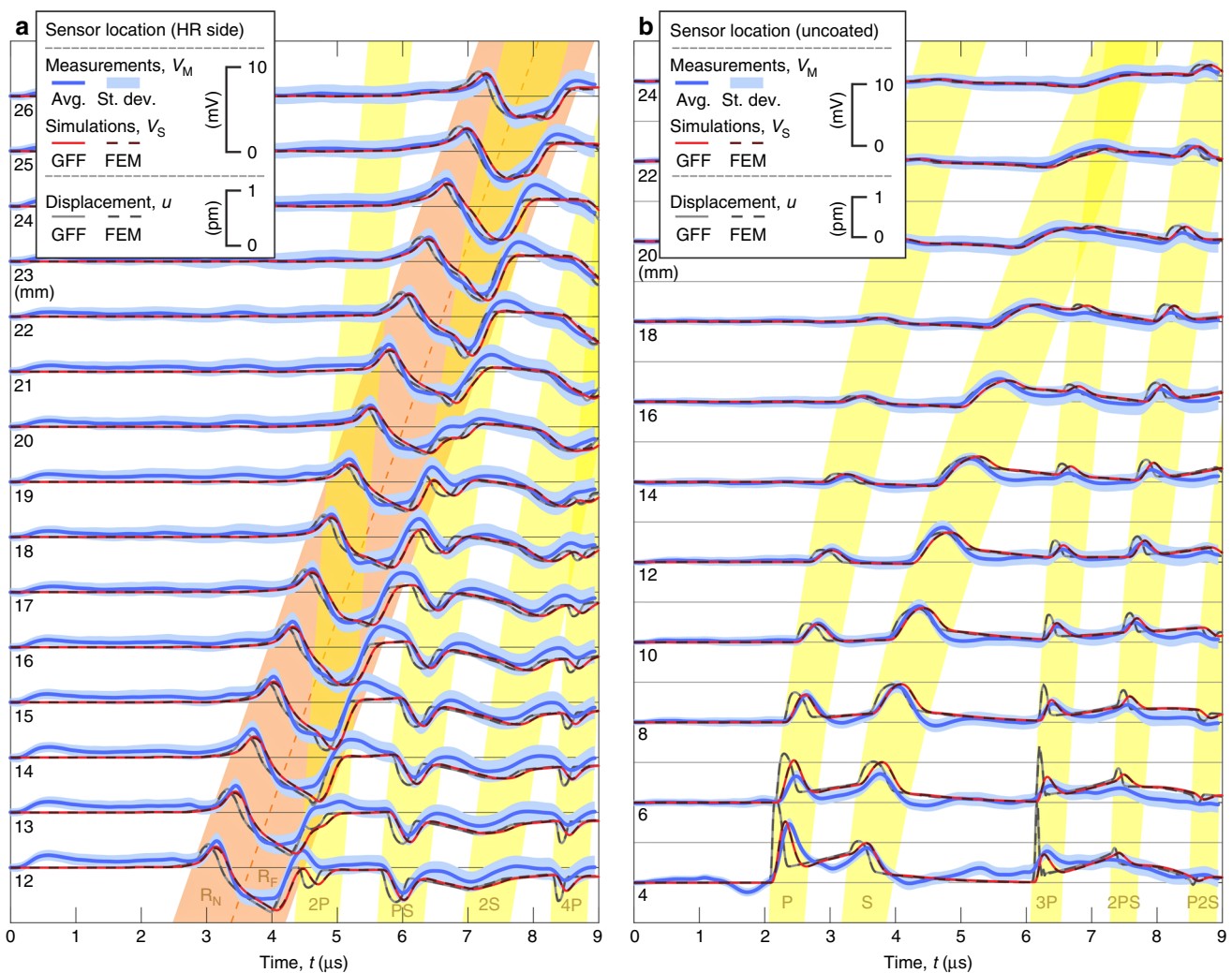

**Fig. 2** Experimental and simulated picometer-scale normal surface displacements. The normal surface displacements are caused by the reverberations of radiation-pressure-driven elastic waves within a dielectric mirror. **a** The waveforms obtained on the illuminated top side of substrate and **b** on the bottom side of the substrate. The coloured bands identify different wave-types in the measured signals. The yellow bands correspond to the multiply reflected bulk waves (P, S, PS, …), while the orange band corresponds to surface-bound Rayleigh wave (R). Contributions to the Rayleigh wave from the near edge ($R_N$) and far edge ($R_F$) of the illuminated area, with respect to the sensor, are resolved. The sensor input $u$ is the normal displacement averaged over the aperture of the PZT sensor. The measured sensor output $V_M$ and the simulated output $V_S$ using two different simulation approaches, finite element modelling (FEM) and Green's function formalism (GFF), show excellent agreement. The modelled waveforms are calculated by considering only radiation pressure as a source of elastic waves and excluding thermoelastic expansion and radial stress. The close match between the measurement and simulations provides clear evidence that the measured surface displacement field is caused predominantly by radiation pressure

(AB), Minkowski (MN), Einstein-Laub (EL), Chu, and Amperian (AMP). While all formalisms produce the same surface-projected normal force distributions, which are correlated directly to the spatial distribution of the illumination intensity, they produce different shear force distributions in the outer area of the edge of the illumination intensity spot, as seen in Fig. 3. This discrepancy is important, as it implies that it can be possible to empirically validate an electrodynamic formalism through sufficiently precise measurement of the elastic waveforms generated by optical shear forces.

Optical forces exerted by the pulse produce emanating elastic transients in the dielectric mirror. The creation and propagation of these elastic waves are simulated in a medium with thermoelastic properties matching those of the dielectric mirror. The finite element method (FEM) is employed to calculate the evolution of elastic waves from a time-dependent thermoelastic equation with realistic boundary conditions using the FDTD-

calculated surface-projected optical force as an input source (see Methods for more discussion on the FEM method). An animation depicts the time evolution of normal momentum in the substrate (Supplementary Movie), of which six frames are presented in Fig. 4. Surface displacement waveforms as detected by the sensor for each sensor-source arrangement are calculated by averaging the simulated surface displacement under the area of the contact tip employed in the measurements. Corresponding sensor output signals, which can be directly compared to experimental measurements, are synthesized by convolving the average simulated surface displacement with the transfer function of the piezoelectric sensor.

Results from the FEM simulations are further supported by calculations using a statistically enhanced Green's function formalism (GFF)[33,36], which combines statistically weighted Green's functions to incorporate the illumination and detection area distributions and their relative positions into an area-to-area

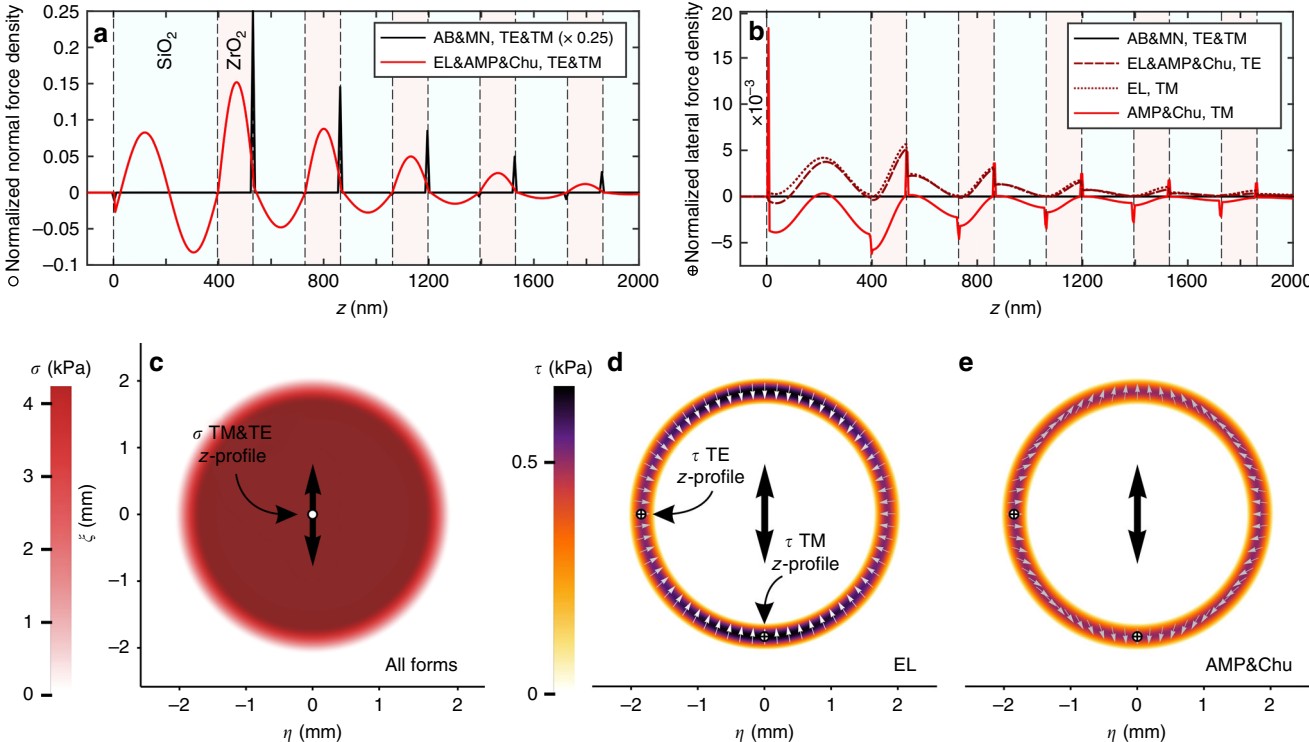

**Fig. 3** Peak force densities and stresses in the HR coating. Simulated light-matter momentum transfers in the HR coating layers and time-average peak normal $\sigma$ and lateral stress $\tau$ acting on the illuminated surface of the mirror. **a** The calculated $z$-distribution of the peak normal force density along the beam axis for all formalisms and both, the TE and the TM polarizations. The AB and MN force densities are degenerate and localized at interfaces, where a push is predicted at the interface of $ZrO_2/SiO_2$, and a pull at the boundary of $SiO_2/ZrO_2$. The EL, AMP, and Chu force densities are also degenerate and continuously distributed throughout the bulk. **b** shows the $z$-profiles of the peak lateral force density within the grating for the TE and the TM illumination at the corresponding maximum slopes of the beam's spatial profile. In addition to the continuously distributed lateral force density throughout the bulk, the degenerate AMP and Chu formulations also yield interface contributions. The presented force densities are normalized by the maximum value calculated by the AB and MN formalism. Utilizing such $z$-profiles, the surface-projected normal $\sigma$ (**c**) and lateral stresses $\tau$ (**d**, **e**) are constructed and later used as source terms in the FEM and GFF wave propagation analysis. The spatial distribution of the normal stress (radiation pressure) has the same shape as the spatial profile of the laser beam and is independent of the polarization and formalism used. Its peak value is 4.24 kPa. The lateral stress-electrostriction has a nonzero value only at points in the laser beam profile where the illumination transitions down from the central uniform plateau. It is dependent both on the direction of polarization (the large arrow marks the oscillation direction of the electric field) and on the chosen electrodynamic formalism. Perpendicularly to the direction of polarization (TE), the maximum magnitude of the shear stress is 0.51 kPa (EL, AMP, and Chu), while along the direction of polarization (TM), it is 0.50 kPa (AMP and Chu) and 0.67 kPa (EL). AB and MN formalisms do not account for electrostriction forces. The EL lateral stress causes the toothpaste-tube effect. The total force exerted by a laser beam pulse on the mirror is independent of the choice of electrodynamic formulation

transfer function of the substrate. In addition to validating FEM simulation results, GFF simulations are used for two purposes. First, description of elastic wave propagation using generalized ray theory enables features in the displacement waveforms and sensor signals to be correlated to different types of elastic waves emanating from the illuminated region. Second, GFF is used also to calculate the sensor signals due to a fully 3D nonaxisymmetric lateral force density distributions (see Methods for more discussion on the GFF method).

Figure 2 shows sensor signals simulated by FEM and GFF methods assuming elastic waves driven solely by radiation pressure. The two simulation results are nearly identical, with slight discrepancies due to numerical artifacts inherent to the FEM method. Close match of the simulated results to the experimental measurements—in terms of amplitude and temporal variation—suggests that measured elastic waves are predominantly driven by the momentum of the incident light pulse. Statistical analysis (see Methods) reveals that a slightly greater fit between measured and simulated signals can be achieved if the simulations incorporate thermoelastic

contributions, although these minor corrections are near the noise levels of the experimental measurement. In Figs. 2, 4, and 5, prominent features in the measured and simulated waveforms are identified as arising from either primary (P), secondary (S), mode converted (combination of the two, such as the PS-wave), head (H), and Rayleigh (R) elastic waves, where the corresponding numbers denote the number of passes that each wave type has made through the bulk of the substrate (see Methods for further details).

Figure 5 compares the measured displacement waveforms with the GFF-simulated waveforms caused separately by radiation pressure $u_z^\sigma$, thermal absorption $u_z^{\mathrm{TEW}}$ considering all the nonreflected laser pulse energy (0.07%) is absorbed, or shear optical forces $u_z^\tau$ for two polarization directions using the Chu and Amperian formalisms (see Methods for detailed description). The measured waveform features closely match the predicted waveform features due to radiation pressure, underscoring the dominant role of momentum transfer in the experimental configuration, but are inconsistent both in amplitude and temporal-variation with predicted waveform features due to

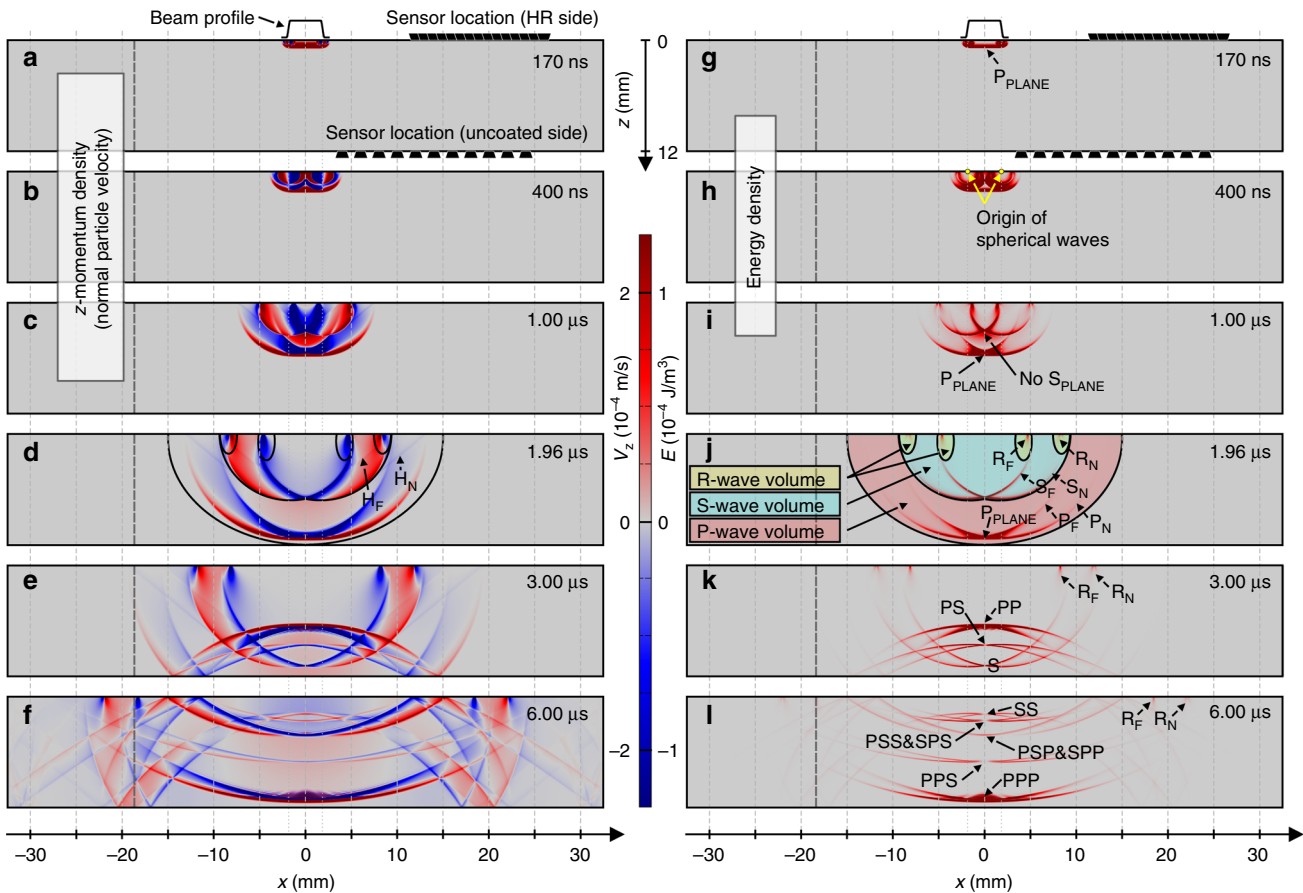

**Fig. 4** Simulated propagation of radiation-pressure-driven elastic waves within a mirror. Six time frames reveal the temporal evolution of **a**–**f** the normal component of particle velocity field $v_z$, which is directly proportional to the normal component of momentum density, and **g**–**l** the mechanical energy density. The elastodynamic picture describes how $z$-momentum and energy are localized in various wave-types, each propagating with a different velocity. In the first snapshot (**a**, **g**), taken at the end of illumination, the dominant disturbance is the plane P-wave. In the second (**b**, **h**) and third (**c**, **i**) snapshots, multicomponent waves originate from the edge of the interaction area and overlap. In the fourth snapshot (**d**, **j**), taken just before the P-wave reaches the bottom facet, these multicomponent waves decouple. In the fifth snapshot (**e**, **k**), reflection and mode conversion of the P-wave is evident. In the sixth snapshot (**f**, **l**), the propagation and reflection of multiple elastic waves give rise to a complex pattern of mechanical disturbances. We use the fourth snapshot (**d**, **j**), to divide the disturbed volume into cylindrically symmetric subvolumes to categorize different wave-types and the resulting surface displacement at the sensor positions shown in black. Through this classification, we estimate that the P-waves carry 86%, S-waves 11%, and R-waves 3% of the total mechanical energy of 1.96 fJ. Interestingly, the transferred $z$-momentum of +1.07 nNs equals twice the incident momentum of the light pulse and is mainly transported by the S-waves (+105%), while the negative $z$-momentum transported by the R-waves (−32%) is partially compensated by the positive $z$-momentum transported by the P-waves (+26%). A time evolution animation of the surface displacement and elastic waves within the mirror can be seen in the Supplementary Movie

thermal absorption. Thus, the shape of elastic waves generated on flat substrates can be used to distinguish radiation pressure and thermal effects. In specially designed microcantilever experiments, the radiation pressure and the thermoelastic contributions were similarly distinguished based on the direction of motion, e.g., by the bending direction of the cantilever[37,38]. Interestingly, shear optical forces predicted for p- and s-polarized light using the Chu and Amperian formulations give rise to minute surface displacement, with waveform amplitudes more than two orders of magnitude smaller than the measured ones.

## Discussion

We have measured picometer-scale surface displacements on a dielectric mirror illuminated by an incident light pulse, under experimental conditions designed to mitigate absorption. Ab initio simulations of momentum deposition and material deformation yield simulated waveforms that closely match the

experimental measurement, confirming that the measured surface displacements are driven almost entirely by the momentum of light.

The simulation platform enables spatio-temporal tracking of energy and momentum distribution in arbitrary configurations and the identification of different elastic wave types generated by light–matter interaction. We foresee potential use of these methods for materials characterization, as optically induced elastic waves provide unique signatures dependent on local optical and viscoelastic properties.

In general, electrodynamic formalisms differ in their predictions of time-averaged optical force distribution[20,22,39,40], which can potentially be empirically validated by resolving the features of minute elastic waves driven by these forces. For the case of illumination of a nonmagnetic dielectric mirror, differences among the various electrodynamic theories are present in the small lateral component of the surface-projected force density (Fig. 3d, f) and the corresponding material displacements (the

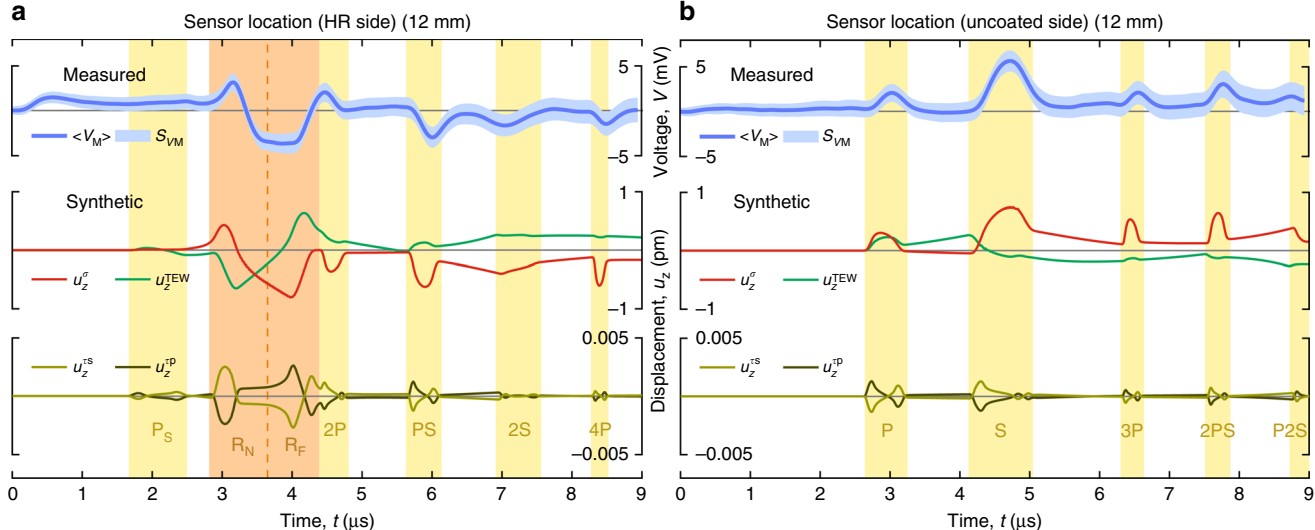

**Fig. 5** Comparison between simulated and measured displacement waveforms. The averaged sensor signal $\langle V_M \rangle$, including the standard deviation $S_{VM}$ at 12 mm beam-center to sensor-center separation distance on the illuminated side of the mirror (**a**) and on the opposite side (**b**) is compared to the calculated synthetic displacements when driven exclusively by normal optical stress (radiation pressure) $u_z^\sigma$; solely by thermal absorption $u_z^{TEW}$ if all of the nonreflected light is assumed to be absorbed in the coating; or only by radial optical shear stress (electrostriction as given by the Chu and AMP formalism) for s-polarization $u_z^{\tau s}$ and p-polarization $u_z^{\tau p}$. The features of the measured signal are present in the simulated displacement waveform driven by radiation pressure. Thermoelastic generation yields displacement waveforms that do not match those in the measured signal. Polarization- and EM-formalism-dependent electrostriction yields displacement waveforms with amplitudes that are more than two orders of magnitude smaller, with a shape that does not match the measured signal. These comparisons further suggest that the surface displacement measurements are driven almost exclusively by normal radiation pressure, as opposed to absorption or shear optical forces

bottom row in Fig. 5). One way to isolate these lateral force densities is to measure and subtract the detected waveforms from orthogonal polarization states. This removes the dominant contributions of the normal force density component (radiation pressure) and thermoelastic effects and results in a residual polarization-dependent signal driven by the difference between the lateral components of the surface-projected force densities (electrostriction). Comparing this residual signal to predictions from the numerical routine developed in this work enables empirical validation of an electrodynamic formalism. The current experimental setup does not have sufficient sensitivity for these measurements, as the displacement-amplitude of the subtracted signal is on the order of 10 fm, below the current noise level of about 40 fm. Further improvements in detection sensitivity, such as the use of HR coatings with larger LIDT and better low-loss coating materials such as $Ta_2O_5/SiO_2$ or $HfO_2/SiO_2$, could bring the signal-to-noise ratio to a level where residual signal detection would be possible. In addition, further refinement of the current approach described here could be achieved by using substrates with no coatings (bare), HR coatings and antireflective coatings, with all possible combinations on the two facets of the ultra low-loss substrates such as Suprasil 300 or Corning 7980. Using this approach, comparative experiments could isolate differences amongst various formalisms based on the normal component of the time-averaged force density, which generates detectable displacement-amplitudes in the range of a few 100 fm.

## Methods

**Substrate**. The elastic mirror (Altechna, custom made) is a cylinder of radius $R_m = 25.4$ mm and thickness $L = 12.0$ mm made out of fused silica $SiO_2$ and coated with HR film on its top surface. Its elastic material properties are: mass density $\rho = 2200$ kg m$^{-3}$, Young's modulus $Y = 72$ GPa, and Poisson's ratio $\nu = 0.17$, while its thermal material properties are: linear thermal expansion coefficient $\alpha_T = 5.6 \times 10^{-7}$ K$^{-1}$, thermal conductivity $k = 13.8$ W m$^{-1}$ K$^{-1}$, and specific heat $C_p = 700$ J kg$^{-1}$ K$^{-1}$. From them additional parameters were calculated: Grüneisen parameter $\gamma = 0.0397$, pressure wave propagation velocity $c_P = 5931.0$ m s$^{-1}$, shear wave

propagation velocity $c_S = 3739.8$ m s$^{-1}$, and Rayleigh wave propagation velocity $c_R = 3387.4$ m s$^{-1}$.

**Coating**. The HR coating (Optida, custom made) functions as a Bragg mirror as it is composed of alternating layers of $ZrO_2$ and $SiO_2$. They are simulated as lossless dielectrics with refractive indices of $n_z = 1.883$ for $ZrO_2$ and $n_s = 1.434$ for $SiO_2$ at the laser pulse wavelength of 1064 nm. The top layer of the coating is a 382.78 nm thick $SiO_2$ layer with alternating layers of 136.93 nm thick $ZrO_2$ and 191.39 nm thick $SiO_2$. The total thickness of the coated substrate is 5444.51 nm. The LIDT fluence is 10 J cm$^{-2}$. A reflectivity of $R = 99.93\%$ is measured for normally incident light at 1064 nm. The coating is sufficiently thin that it does not affect the propagation of elastic waves in the substrate. All values are reported by the manufacturer.

**Laser pulse**. Linearly polarized light pulses are generated with a 0.5 Hz repetition by a Q-switched Nd:YAG laser (Fotona, QX MAX) at the vacuum wavelength of 1064 nm. The pulses are collimated so that their wavefronts are nearly planar. Their measured intensity is fitted by an analytical function $I(r, t) = E_0 \eta(r) \theta(t)$ with normalized and separable space and time dependencies and a measured energy of $E_0 = (160 \pm 3)$ mJ (Gentec-eo, monitor SOLO-PE; energy detector, QE 50SP-H-MB; attenuator, QEA-50). The spatial distribution $\eta(r)$ is measured (Ophir, BM-USB-SP620 BeamMic) to be approximately a circularly symmetric, nearly top-hat function with an overall radius of 2.05 mm. The spatial distribution is a flat plateau within the inner 4/5 of laser beam pulse radius and a smooth cosine taper to zero in the outer 1/5 of the pulse radius. The time profile $\theta(t)$ is fitted to the measured (photodetector: ALS, MSM-Photodetector Series, 20 GHz; oscilloscope: Keysight/Agilent, DSO81204B Infiniium Oscilloscope, 12 GHz) cycle-averaged envelope, which consists of an asymmetric bell with a fast rise-time and a more gently decreasing tail. The pulse has a duration of 170 ns, a peak at 50 ns, and a FWHM of 19 ns within which 64% of the total energy resides. The maximum intensity of the pulse is 650 GW m$^{-2}$ which gives, upon reflection from a perfect mirror, a maximum radiation pressure of 4.3 kPa. The average laser pulse power during its duration is 0.94 MW, while its peak power is 6.9 MW. The maximum fluence of the laser pulse is 1.5 J cm$^{-2}$, which is well below the reported LIDT of the substrate. To block Rayleigh scattering in the surrounding air atmosphere, the light was delivered through an opaque black tube.

**Sensor**. The piezoelectric sensor (KRN Services, KRNBB-PC) consists of a conically shaped PZT-5A crystal with a circular contact tip with a radius of $R_0 = 0.5$ mm and an assumed uniform sensitivity in its sensing area. It is designed to measure absolute out-of-plane displacements a few orders of magnitude around a

**Table 1 Formulations of electrodynamics**

| Formulation | Momentum density G | Stress tensor $\overline{\overline{\mathbf{T}}}$ |
|---|---|---|
| Minkowski | $\mathbf{D} \times \mathbf{B}$ | $(\mathbf{D} \cdot \mathbf{E} + \mathbf{B} \cdot \mathbf{H})\overline{\overline{\mathbf{I}}}/2 - \mathbf{DE} - \mathbf{BH}$ |
| Abraham | $(\mathbf{E} \times \mathbf{H})/c^2$ | $\left[(\mathbf{D} \cdot \mathbf{E} + \mathbf{B} \cdot \mathbf{H})\overline{\overline{\mathbf{I}}} - \mathbf{DE} - \mathbf{BH} - \mathbf{ED} - \mathbf{HB}\right]/2$ |
| Einstein-Laub | $(\mathbf{E} \times \mathbf{H})/c^2$ | $(\varepsilon_0 E^2 + \mu_0 H^2)\overline{\overline{\mathbf{I}}}/2 - \mathbf{DE} - \mathbf{BH}$ |
| Amperian | $\varepsilon_0(\mathbf{E} \times \mathbf{B})$ | $(\varepsilon_0 E^2 + \mu_0^{-1}B^2)\overline{\overline{\mathbf{I}}}/2 - \varepsilon_0\mathbf{EE} - \mu_0^{-1}\mathbf{BB}$ |
| Chu | $(\mathbf{E} \times \mathbf{H})/c^2$ | $(\varepsilon_0 E^2 + \mu_0 H^2)\overline{\overline{\mathbf{I}}}/2 - \varepsilon_0\mathbf{EE} - \mu_0\mathbf{HH}$ |

The electromagnetic field momentum density and stress tensor for each electrodynamic formulation as reported in the literature[17,21]. The stress tensor terms are expanded as dyadics

picometer with a reasonably flat frequency response between 10 kHz and 5 MHz while minimizing ultrasonic and electromagnetic distortion effects. The sensor is calibrated beforehand and its transfer function is known[34]. Due to its finite aperture, all detected displacements are spatially averaged by the sensor. The tip is covered by a protective nickel foil, but due to its inadequate reflectivity ($R = 73\%$ at 1064 nm) a soft gold foil ($R = 99\%$ at 1064 nm, thickness: 20 μm) was inserted between the sensor tip and the substrate to further suppress absorption of the scattered laser pulse and to provide a good contact. The head of the sensor was vacuum gold coated to further increase the reflectance of the stray light.

**Setup geometry**. The laser pulse arrives from air and normally illuminates the top surface (HR-coated) of the substrate at various positions. The origin of the Cartesian coordinate system $(x, y, z)$ is defined by the intersection of the top surface and the symmetry axis of the laser beam and depends on the position of the laser pulse reflection. The $x$–$y$ plane coincides with the top surface of the substrate at $z = 0$ and the $z$ axis is directed into the substrate. The bottom (uncoated) surface of the substrate lies in the $z = L$ plane. Alternatively, a cylindrical coordinate system $(r, \varphi, z)$ could be introduced with $r$ being the radial distance and $\varphi$ the azimuth. The detector is deployed at $y = 0$ either on the top or the bottom surface with the center of the contact tip each time displaced by $\Delta x = 7$ mm from the center of the facet. While it remained fixed on one of the surfaces, the distance $x_i$ between the centers of the illuminated spots and the sensor tip is changed between the sets of measurements and with it, the origin of the coordinates changes as well. When the sensor is deployed on the top surface at $(x_i, 0, 0)$, 15 lateral arrangements were chosen from $x_1 = 12$ mm to $x_{15} = 26$ mm in 1-mm increments. When it was deployed on the bottom surface at $(x_i, 0, L)$, 11 lateral arrangements were made extending from $x_1 = 4$ mm to $x_{11} = 24$ mm in 2-mm increments.

**Polarization**. Based on the experimental setup, the $x$–$z$ plane, which includes the sensor position, is defined as the plane of incidence. Linearly polarized incident light is referenced as p-polarized if its electric field oscillates parallel to the plane of incidence ($x$–$z$ plane) and s-polarized if its electric field oscillates perpendicular to the plane of incidence ($y$–$z$ plane).

**Statistical analysis**. Two hundred time-dependent measurements for each of the 26 sensor arrangements are averaged, their standard deviation are calculated, and Pearson's $\chi^2$-test is performed with 31 degrees of freedom to determine whether the accompanying noise was normally distributed and therefore stochastic. To quantify the match between the synthetic waveforms and averaged measurements, the average distance and standard deviation between them is calculated.

**Formulations of electrodynamics**. The five prevalent electrodynamic formulations considered in this work are: Abraham (AB), Minkowski (MN), Einstein-Laub (EL), Chu, and Amperian (AMP). Each of these have been shown to obey the conservations of energy and momentum[21,41]. Using these formulations, the dynamic interaction between matter and the electromagnetic field can be described using the momentum continuity equation,

$$\mathbf{f} = -\nabla \cdot \overline{\overline{\mathbf{T}}} - \frac{\partial}{\partial t}\mathbf{G}, \tag{1}$$

where, $\mathbf{f}$ is the force density, $\mathbf{G}$ is the electromagnetic momentum density, and $\overline{\overline{\mathbf{T}}}$ is the electromagnetic stress tensor. The values of $\mathbf{G}$ and $\overline{\overline{\mathbf{T}}}$ for each formulation are listed in Table 1 with $\mathbf{f}$ following from them. In Table 1, $\mathbf{E}$ is the electric field, $\mathbf{H}$ is the magnetic field, $\mathbf{D}$ is the electric displacement field, $\mathbf{B}$ is the magnetic flux density, $\varepsilon_0$ is the free-space permittivity, $\mu_0$ is the free-space permeability, $c$ is the speed of light in vacuum, and $\overline{\overline{\mathbf{I}}}$ is the identity matrix.

**Force density calculations (FDTD method)**. Assuming a linearly polarized light pulse, another Cartesian coordinate system $(\xi, \eta, z)$, rotated around the $z$-axis, is

introduced, where the $\xi$–$z$ plane is always parallel to the electric field vector $\mathbf{E}(\xi, z)$ at any point and thus follows the direction of its polarization.

For each electromagnetic formulation, Eq. (1) along with the boundary conditions within the layered HR coating is solved numerically using the FDTD method in 2D. Two 2D slices of the actual 3D interaction volume are simulated. The 2D TM simulation (nonzero components are: $E_\xi$, $E_z$, and $H_\eta$) computes the force density within the central slice of the 3D interaction volume in the $\xi$–$z$ plane and the TE simulation (nonzero components are: $H_\eta$, $H_z$, and $E_\xi$) computes it in the $\eta$–$z$ plane. These two separate 2D simulations are used in the angular interpolation to reconstruct the full 3D force density, approximating the results obtained directly by a full 3D model[20].

Note that p- and s-polarization are reserved for the experimental direction of linear polarization with respect to the position of the sensor while TM and TE modes are used to describe the 2D slices with respect to the direction of linear polarization.

The force density is solved by discretizing Eq. (1). For example, the $z$-component of the force density for the TM polarization, is numerically solved for via:

$$f_z{\binom{i}{j+1/2}}^{n+1/2} = -\left[\frac{1}{\Delta\xi}\left(T_{z\xi}{\binom{i+1/2}{j+1/2}}^{n+1/2} - T_{z\xi}{\binom{i-1/2}{j+1/2}}^{n+1/2}\right) + \frac{1}{\Delta z}\left(T_{zz}{\binom{i}{j+1}}^{n+1/2} - T_{zz}{\binom{i}{j}}^{n+1/2}\right)\right]$$
$$- \frac{1}{\Delta t}\left[G_z{\binom{i}{j+1/2}}^{n+1} - G_z{\binom{i}{j+1/2}}^{n}\right], \tag{2}$$

where, time is indexed by $n$, the lateral dimension in the 2D plane (which is $\eta$ for TE, and $\xi$ for TM) is indexed by $i$, the $z$-direction is indexed by $j$, and $N_i$ with $N_j$ represent every $\Delta\xi\Delta z$-large discrete cell of the coating slice. $T_{z\xi}$ and $T_{zz}$ are the components of the stress tensor, solved from FDTD field values using field averaging as described in the literature[21].

Within the 2D slice and at each time instant, the total cycle-averaged, normal force generated by the pulse on the coating is

$$F_z^{n+1/2} = \sum_i^{N_i}\sum_j^{N_j} f_z{\binom{i}{j+1/2}}^{n+1/2}\Delta\xi\Delta z, \tag{3}$$

obtained by summing the cycle-averaged force density distribution over the entire interaction volume within the coating slice. As this quantity is independent of the chosen formalism, the cycle-averaged, surface-projected, normal force

$$f_{z(i)}^{n+1/2} = \sum_j^{N_j} f_z{\binom{i}{j+1/2}}^{n+1/2}\Delta z, \tag{4}$$

in general depends on the form of the momentum continuity equation.

The lateral force is solved in a similar fashion. In the lateral direction, the actual beam width cannot be directly simulated with reasonably available computational power, as a spatial step size of ~3 nm would be needed over a width of 4 mm. Instead the lateral results are calculated for a smaller width, with beam parameters also scaled, and then normalized to represent the actual width, as is common in other numerical methods such as fluid dynamics where this issue is more commonly encountered[42]. The total cycle-averaged, lateral force generated by the pulse on the coating is always 0, while the surface-projected distribution again depends on the choice of the formalism.

The input quantity for the wave propagation and waveform simulations is the surface-projected 3D distribution of the force density. This 2D field residing in the $x$–$y$ plane has a normal component that corresponds to normal pressure ($\sigma$, radiation pressure) and a lateral component corresponding to shear stress ($\tau$, electrostriction).

**Wave propagation modelling (FEM).** Optical forces in addition to thermal effects induce surface displacements of the sample. The displacement field $\mathbf{u}(r, z, t \geq 0)$ is calculated by solving the uncoupled time-dependent thermoelastic equation, Eq. (5), along with the heat conduction equation, Eq. (6), with appropriate boundary (all stress components are zero on all boundary surfaces) and quiescent initial conditions, $\mathbf{u}(r, z, 0)$ and $\partial \mathbf{u}(r, z, t)/\partial t|_{t=0}$. The FEM was applied for the numerical calculations using Comsol Multiphysics 4.3b software to solve the following equations:

$$(1 - 2\nu)\nabla^2 \mathbf{u}(r, z, t) + \nabla[\nabla \cdot \mathbf{u}(r, z, t)] =$$
$$\frac{2(1+\nu)(1-2\nu)}{Y}\left[\rho \frac{\partial^2 \mathbf{u}(r,z,t)}{\partial t^2} + \mathbf{F}_V\right] + 2(1+\nu)\alpha_T \nabla T(r, z, t), \quad (5)$$

$$\begin{cases} \rho C_p \dfrac{\partial T(r,z,t)}{\partial t} - k\nabla^2 T(r, z, t) = 0 \\ -\mathbf{n} \cdot [-k\nabla T(r, z, t)]\Big|_{z=0} = Q_B(r, z = 0, t). \end{cases} \quad (6)$$

$T(r, z, t)$ is the temperature change upon laser excitation due to absorption, $Q_B(r, z = 0, t) = (1 - R)I(r, t)$ is the boundary heat source, and $I(r, t)$ is the measured laser pulse intensity, $\mathbf{F}_V$ is the body load defined as force per unit volume. Alternatively, a boundary load applied to the surface can be described as $\sigma \cdot \mathbf{n} = \mathbf{F}_A$. Here, $\mathbf{F}_A = F_{A,r}\hat{\mathbf{r}} + F_{A,z}\hat{\mathbf{z}}$ is the boundary load defined as force per unit area and $\sigma$ represents the stress components. $F_{A,r}$ and $F_{A,z}$ are the radial and normal components of the boundary load, respectively. $\mathbf{n}$, $\hat{\mathbf{r}}$, and $\hat{\mathbf{z}}$ are the normal to the selected surface, radial and normal unit vectors, respectively. The linearity of the problem allows for the separate calculation of thermoelastic waves (TEWs) and those induced separately by the normal and lateral component of optical forces.

The TEWs are calculated assuming $F_{A,r} = F_{A,z} = 0$ while the second term in the square brackets on the right-hand side of Eq. (5) is not considered. Surface displacement was calculated by solving Eqs. (5) and (6) using the real physical properties of the substrate $Y$, $\rho$, $\nu$, $\alpha_T$, $k$, and $C_p$.

Waveforms due to normal and lateral optical forces are calculated assuming negligible light absorption by the sample, which implies that the last term on the right-hand side of Eq. (5) is not considered. Surface displacement is calculated by solving Eq. (5) with the optical forces obtained by FDTD calculations. The normal optical force is circularly symmetric and indistinguishable for all the different electromagnetic formalisms and linear polarization directions. It has the expected form $F_{A,z} = 2I(r, t)/c$ for radiation pressure acting during the reflection of a light pulse from a perfect mirror. On the other hand, the radial force density is in general not circularly symmetric and depends on the formalism used and on the direction of linear polarization. Nonsymmetric force distribution requires a 3D description of the problem, which is attained with Green's function approach.

The FEM model was built in the 2D axisymmetric geometry assuming that the excitation laser is off-centered by $\Delta x$. For simplicity, the sample radius is considered to be $a = R_m + \Delta x = 32.4$ mm. The realistic geometric limits of the sample are depicted in the figures. The normal component of the surface displacement is computed for the top $u_z(r, z = 0, t)$ and bottom $u_z(r, z = L, t)$ surface of the sample and the results are used to calculate the average displacement under the tip of the sensor.

**Waveform modelling (GFF).** The statistically enhanced GFF method is used to simulate the light-induced (thermo)elastic source of ultrasonic waves and their subsequent propagation and reverberation in the substrate[33]. It solves similar thermoelastic equations as given in Eqs. (5) and (6) with the same initial and boundary conditions and provides waveforms (time evolutions) a detector might sense. Due to short simulation time-frames, the thermal conductivity was set to zero ($k = 0$) modelling a diffusionless energy built up within a thin surface layer of thickness $d$ with the temperature field of:

$$T(r, z = 0, t) = \frac{(1 - R)E_0}{\rho d C_p}\eta(r)\int_0^t \theta(\tau)d\tau. \quad (7)$$

Algebraic linearity is assumed throughout the modelling process, allowing for the use of superposition and convolution operations.

Solutions are obtained through the generalized ray theory using methods developed by Hsu[36] for a plate and are partly modified by those calculated by Kausel[43] for a semi-infinite space. The building blocks of such solutions are the material-specific directional Green's point-to-point transfer functions which transfer a $\delta$-force input signal in the direction $j$ at an input point $\mathbf{v}_0$ to an output displacement signal in the direction $i$ at an output point $\mathbf{w}_0$. As $\mathbf{v}_0$ and $\mathbf{w}_0$ represent the centers of illuminated and detection areas, respectively, the $x$-axis coincides with $\mathbf{v}_0 - \mathbf{w}_0$.

To obtain the material displacement waveforms, there are two relevant types of Green's functions that correspond to two main ultrasound-inducing mechanisms during the light–matter interaction. Optical forces require two-indexed Green's functions $g_{ij}(\mathbf{v}_0, \mathbf{w}_0, t)$ to transfer the light momentum $\Delta p_j$ to the source volume, while the thermal expansion due to light absorption requires their spatial derivatives, the three-indexed Green's functions $g_{ij,k} = \partial g_{ij}/\partial x_k$, to transfer isotropic

force dipoles with magnitudes of $D = \gamma(1 - R)E_0$. Since only the out-of-plane displacements $u_z(\mathbf{w}_0, t)$ are of interest, the required set is reduced to $g_{zx}$, $g_{zy}$, and $g_{zz}$ among pressure-transferring Green's functions and to $g_{zx,x}$, $g_{zy,y}$, and $g_{zz,z}$ among expansion-transferring Green's functions.

The general $z$-displacement induced by an optical point source is expressed in terms of Green's functions as:

$$u_z(\mathbf{w}_0, t) = \left[\sum_{j=x,z}\Delta p_j g_{zj}(\mathbf{v}_0, \mathbf{w}_0, t) + D\sum_{j=x,y}h_{zj,j}(\mathbf{v}_0, \mathbf{w}_0, t)\right] * \theta(t). \quad (8)$$

To represent the heat deposition, the accumulating Green's functions $h_{zj,j}$ are introduced as time-convolutions of the impulse functions with a Heaviside step function: $h_{zj,j}(t) \equiv g_{zj,j}(t) * H(t)$. Temporal convolution, designated by the asterisk operator, is also used to incorporate the temporal profile of the light pulse into a displacement waveform.

The detected waveforms are additionally affected by the temporal and areal distributions of the surface-projected force densities and absorption profiles in the illuminated area as well as by the sensor frequency- and area-distributed sensitivity over its contact area. In general, areal distributions of both interactions and their relative positions are incorporated, by means of geometric probability[44], into a normalized area-to-area weight function $\psi(\mathbf{v}, \mathbf{w})$. This is combined with a continuous Green's function field $g(\mathbf{v}, \mathbf{w}, t)$, comprised of a multitude of neighboring Green's functions, to form the area-to-area transfer function $\int_{-\infty}^{\infty}\psi(\mathbf{v}, \mathbf{w})\,g(\mathbf{v}, \mathbf{w}, t)\,d\|\mathbf{v} - \mathbf{w}\|$[33]. From it, the general displacement waveforms are calculated as

$$u(\mathbf{w}, t) = J\left[\int_{-\infty}^{\infty}\psi(\mathbf{v}, \mathbf{w})\,g(\mathbf{v}, \mathbf{w}, t)\,d\|\mathbf{v} - \mathbf{w}\|\right] * \theta(t). \quad (9)$$

The calculated input for the sensor $u_z = u_z^{\sigma} + u_z^{\tau} + u_z^{TEW}$ is a linear superposition of three separate types of displacement waveforms, averaged over the sensor aperture.

The waveforms due to the normal component of optical forces (radiation pressure) $u_z^{\sigma}$ and the thermoelastic waveforms $u_z^{TEW}$ are polarization and formalism independent. If the force impulse acts perpendicularly to the $x$-axis in such a lateral arrangement, it does not induce any displacement in the $z$-direction, making $g_{zy} = 0$. Since the thermal part of the source is on the surface, $g_{zz,z} = 0$ as well. Their waveforms are calculated as:

$$u_z^{\sigma}(\mathbf{w}, t) = J_{\sigma}\left[\int_{-\infty}^{\infty}\psi_{\sigma}(\mathbf{v}, \mathbf{w})\,g_{zz}(\mathbf{v}, \mathbf{w}, t)\,d\|\mathbf{v} - \mathbf{w}\|\right] * \theta(t), \quad (10)$$

$$u_z^{TEW}(\mathbf{w}, t) = D\left[\int_{-\infty}^{\infty}\psi_{\sigma}(\mathbf{v}, \mathbf{w})\Big(h_{zx,x}(\mathbf{v}, \mathbf{w}, t) + h_{zy,y}(\mathbf{v}, \mathbf{w}, t)\Big)\,d\|\mathbf{v} - \mathbf{w}\|\right] * \theta(t). \quad (11)$$

$J_{\sigma} = 2E_0/c$ is the magnitude of the normal force impulse that equals twice the momentum of the light pulse.

The waveforms due to the lateral components of optical forces (electrostriction-like effect) $u_z^{\tau}$ in general depend on the orientation of the linearly polarized light and the choice of the electromagnetic formulation. If the incident light is polarized with its electric field parallel to the plane of incidence (the $x$–$z$ plane), the computed waveform is termed $u_z^{\tau P}$ and if it oscillates in the $y$–$z$ plane, the waveform is $u_z^{\tau S}$. Separate contributions are calculated as:

$$u_z^{\tau P}(\mathbf{w}, t) = J_{\tau}\left[\int_{-\infty}^{\infty}\text{proj}_{\|}\Big(\psi_{\tau P}(\mathbf{v}, \mathbf{w})\Big)g_{zx}(\mathbf{v}, \mathbf{w}, t)\,d\|\mathbf{v} - \mathbf{w}\|\right] * \theta(t), \quad (12)$$

$$u_z^{\tau S}(\mathbf{w}, t) = J_{\tau}\left[\int_{-\infty}^{\infty}\text{proj}_{\|}\Big(\psi_{\tau S}(\mathbf{v}, \mathbf{w})\Big)g_{zx}(\mathbf{v}, \mathbf{w}, t)\,d\|\mathbf{v} - \mathbf{w}\|\right] * \theta(t). \quad (13)$$

$J_{\tau}$ is the sum of the magnitudes of infinitesimal lateral force impulses distributed within the interaction volume. The operator $\text{proj}_{\|}()$ stands for the projection of the lateral force vectors onto the line connecting points from each of the two areas.

The sensor input waveform $u_z(t)$ is additionally convolved with the sensor transfer function $s(t)$, describing calibration characteristics, to simulate the measured sensor signal $V(t) = u_z(t) * s(t)$[34].

**Wave-type classification.** The elastic waves reverberating through the substrate are a superposition of several individual transients that are identified as primary (longitudinal) P-waves (P, 2P, 3P, 4P…), secondary (transversal) S-waves (S, 2S…), head H-waves (H), surface Rayleigh R-waves ($R_N$, $R_F$), and combinations of the first two because of mode conversion (PS, 2PS, P2S…). The numbers denote the

number of passes that a specific wave type made through the bulk of the substrate while the indexed letters indicate whether that specific transient originated from the nearer (N) or the farther (F) edge of the beam with regard to the detector. The most convenient way of identifying the wave-type is by its propagation velocity and time of detection.

**Data availability**. The data that support the plots within this paper and other findings of this study are available from the corresponding author upon reasonable request.

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

## Acknowledgements

T.P., J.L., A.B., and R.P. acknowledge the financial support from the Slovenian Research Agency (research core funding No. P2-0270 and P2-0392). G.V.B.L. and N.G.C.A. acknowledge the financial support from the Brazilian agencies CAPES, CNPq, and Fundação Araucária. T.P. and N.G.C.A. acknowledge the support from PVE-CAPES (grant No. 23038.010102/2013-34). K.J.C. acknowledges support from NSERC Discovery Grants program. We thank Altechna for providing the substrate and Optida for the manufacturing of the HR coating.

## Author contributions

T.P. conceived and led the research. T.P. and J.L. developed the wave propagation model utilizing GFF. G.V.B.L. and N.G.C.A. developed the wave propagation model and performed the simulations utilizing FEM. M.B.-W. and K.J.C. developed the interaction model and calculated force densities utilizing FDTD algorithm. J.L. performed waveform simulations utilizing GFF and analyzed the measurements. T.P., A.B., and R.P. developed the experimental setup and performed the measurements. T.P., J.L., K.J.C., and N.G.C.A. wrote the manuscript with inputs and revisions from all authors.
