## [Peer Review File · Nature Communications]

Reviewers' comments:

Reviewer #1 (Remarks to the Author):

When a light pulse hits a mirror it gives it a minute kick - it transfers momentum to the mirror. The present paper reports on a careful experimental and numerical study of how this initial kick reverberates in vibrations inside the material of the mirror. Comparing the measurements with simulations one can draw conclusions about the momentum of light inside dielectric materials, potentially resolving the notorious controversy on the momentum of light. In fact, the authors could experimentally disprove the Chu and Amperian momentum.

This is definitely work worth publishing in Nature Communications. However, I am concerned about the following few points that should be addressed before acceptance:

1. Why do the simulations for the Abraham and the Minkowski momentum agree in Fig. 3? If the Abraham momentum were the correct momentum of light, one would expect a forward push at an interface from lower to higher refractive index, if the Minkowski momentum were the true one, one would get a pulling force for the same situation, see PRL 101 243601 (2008). Table 1 shows that the expressions for the two contenders are different - so why are the results the same?
2. For the Abraham stress, is not DE equal to ED and BH equal to ED ? This would give the same stress as for the Minkowski case (as it should).
2. It would be useful to discuss PRL 101 243601 (2008) that is related, but performed with much simpler means, and controversial. What does the present study imply?
3. Is the Einstein-Laub momentum not just the Abraham momentum with electrostriction/ dipole force included? See PRA 90, 033801 (2014) for the case of fluids (solids should be similar). One might expect this to be the most promising candidate for explaining the momentum transfer.

Reviewer #2 (Remarks to the Author):

In this manuscript, Pozar et al show modeling and experiments of electromagnetic propagation and the conversion of energy and momentum into elastic waves in a solid material. A clever experimental technique is presented and simulations and experiments are shown to agree well. The results seem plausible and are supported by the data, but it would be helpful if the authors could more clearly state the new results that are presented in this manuscript, with respect to previous works. In the abstract they state that they have "compelling evidence," but they should be more explicit (and quantitative) about what is observed and what is different in this experiment, making it more "compelling" than others. As an example, at the end of the first paragraph the authors state that, elastic waves induced by radiation pressure "have yet to be directly measured." Then the next paragraph begins with "Elastic waves driven by light momentum can be detected as transient normal displacements on the surface of an illuminated object [10–13]," which sounds like they have been measured (see references 12, which is their own work)... I think there are new results in this

manuscript, but those should be more obvious. My further comments are below.

-In the abstract, the authors state that the “details” of the momentum conversion from EM fields to elastic waves within a solid have been predicted by theory, but that it has not been validated by experiments because of heating due to light absorption. There is then very little text within the manuscript describing heating effects on momentum transfer or optomechanical forces. I think this should be discussed within the manuscript to put such effects into context. Two works that discuss and compensate for thermal effects in optical force/momentum experiments are:

(a) Weld et al, Appl. Phys. Lett. 89, 164102 (2006)

(b) Ma et al, Appl. Phys. Lett. 106, 091107 (2015)

The authors should consider adding a short discuss with respect to this.

-The authors mention the Abraham and Minkowski formalisms a few times and discuss, briefly, the controversy resulting from these two descriptions. This is a very interesting point and the authors state that these measurements may shed light on this problem. They say “...it implies that it can be possible to empirically validate an electrodynamic formalism through sufficiently precise measurement of the elastic waveforms generated by optical shear forces.” But, how precise would the measurements need to be? Can they do it? I’d like to see something quantitative here.

-A similar comment appears at the end of the paper: “This work also forms a basis for experimental resolution of the Abraham-Minkowski controversy.” How? What are the different, measureable features? Can this currently be resolved? How far off are the sensitivities? If you are not there yet, what would be needed to get there? These answers should be included in the discussion.

Reviewer #3 (Remarks to the Author):

The work reports on detection of elastic waves generated following the reflection of electromagnetic waves over a non-absorbing dielectric plate due to the transmission of momentum from the field photons into the object. The authors design and develop an original and high sensitive experiment to detect the momentum driven mechanical waves under the condition of mitigated thermal effects. A set a piezoelectric detector conveniently located on the front a rear side of the device are able to detect pm-amplitude mechanical waves. Authors provide convincing data about the reality of the waves. They also use ab-initio electrodynamic models based on different descriptions of the spatial distribution of normal and shear force densities to explain the observed effects. The models describe qualitatively well the observations. Furthermore, authors claim that the proposed experimental method can be used to validate the correct electrodynamic model, although they do not provide a clear evidence of that. The work contains detailed information about the experimental methods and the mathematical models used to describe the electrodynamics of the interaction. The manuscript is well organized and written in good English. Considering the fundamental experiment described and the originality of the contribution, I recommend the manuscript for publication.

Authors' response to Reviewer #1 remarks:

Reviewer #1 remarks are typeset in italic, blue font.

Authors' response follows in upright, black font.

When a light pulse hits a mirror it gives it a minute kick - it transfers momentum to the mirror. The present paper reports on a careful experimental and numerical study of how this initial kick reverberates in vibrations inside the material of the mirror. Comparing the measurements with simulations one can draw conclusions about the momentum of light inside dielectric materials, potentially resolving the notorious controversy on the momentum of light. In fact, the authors could experimentally disprove the Chu and Amperian momentum.

Thank you very much for taking your time to evaluate our work. We have carefully read your constructive comments and prepared the following response. All your suggestions and recommendations have been considered and commented on.

This is definitely work worth publishing in Nature Communications. However, I am concerned about the following few points that should be addressed before acceptance:

Thank you for your positive recommendation. We address your concerns by replying to the points raised below.

1. Why do the simulations for the Abraham and the Minkowski momentum agree in Fig. 3? If the Abraham momentum were the correct momentum of light, one would expect a forward push at an interface from lower to higher refractive index, if the Minkowski momentum were the true one, one would get a pulling force for the same situation, see PRL 101 243601 (2008). Table 1 shows that the expressions for the two contenders are different - so why are the results the same?

We model our experiments using the five leading formulations of classical electromagnetic theory in the quasi-stationary approximation where the medium is not moving. These formulations are: (i) Abraham (AB), (ii) Minkowski (MN), (iii) Einstein-Laub (EL), (iv) Chu (also known as the field-kinetic or EH representation), and (v) Amperian (AMP, also known as the Lorentz, Nelson or EB representation). Within these formulations, there are three different electromagnetic momentum densities: the Livens momentum density $\epsilon_0 \vec{E} \times \vec{B}$ used in the AMP formulation, the Abraham momentum density $\epsilon_0 \mu_0 \vec{E} \times \vec{H}$ used in the AB, EL and Chu formulations, and the Minkowski momentum density $\vec{D} \times \vec{B}$ used in the MN formulation. Here, \vec{E} is the electric field, \vec{H} is the magnetic field, \vec{D} is the electric displacement field, \vec{B} is the magnetic flux density, ϵ_0 is the free-space permittivity, and μ_0 is the free space permeability.

The simulations in Fig. 3 for the Abraham (AB) and Minkowski (MN) formulations agree because the time-averaged (optical cycle-averaged) force densities are the same for both formulations under monochromatic illumination (See Ref. [1.1], Fig. 10a and 10b). However, the instantaneous force density between AB and MN formulations differs (See Ref. [1.1], Figs. 12a). Our mirror is modeled as a piecewise homogeneous and isotropic, non-magnetic, non-dispersive, linear and transparent (lossless) solid. Using these assumptions and employing the constitutive relations $\vec{B} = \mu_0 \vec{H}$ and $\vec{D} = \epsilon_0 \epsilon_r(\vec{r}) \vec{E}$, the time-averaged force density is identically $-\frac{1}{4} \epsilon_0 E(\vec{r})^2 \vec{\nabla} \epsilon_r(\vec{r})$ for both the AB and MN formulations [1.2]. Here, $\epsilon_r(\vec{r}) = n(\vec{r})^2$ is the relative permittivity that equals the square of the refractive index $n(\vec{r})$, \vec{r} is the position vector, and $\vec{\nabla}$ is the gradient operator.

The paper of She, Yu and Feng [1.3] describes a neat experiment to interrogate optical forces in optical fibers. We, however, disagree with their physical interpretation of the results, as do Mansuripur [1.4] and Brevik [1.5] in their comments to [1.3]. As pointed by Mansuripur [1.4] in his third point of the comment, the arguments based solely on the form of momentum density, given by expressions in [1.3] on page 1, 2nd column, lines 1-12, cannot give a correct interpretation of the experiments. The force density that causes material motion includes two terms, as can be seen in Eq. (1) in the Methods section of our paper, where only one term involves momentum density. There is another term related to the stress tensor that has to be taken into account to close the momentum continuity equation between the electromagnetic sub-system and the material sub-system. For that reason, we do not agree with the statement, which is likely derived solely from the momentum density, that *if the Abraham momentum were the correct momentum of light, one would expect a forward push at an interface from lower to higher refractive index, if the Minkowski momentum were the true one, one would get a pulling force for the same situation.*

Our experiments only have access to the time-averaged force density, because we cannot detect material motion at optical frequencies. Our experiments also cannot discern the in-depth distribution of the force density, because it takes place in a depth that is much smaller than the aperture of the sensor (that is why surface-projection was performed to obtain normal and shear stress), but we can detect the lateral distribution, since its extent is larger than the 1-mm aperture radius of the sensor.

The time-averaged force densities for the AB and MN formulations are identical under the above stated assumptions of the model. When the momentum density and stress tensor given in Table 1 are inserted into Eq. (1), the resulting time-averaged force density is identical for AB and MN formulations.

As for the original Abraham-Minkowski debate, this problem has already been theoretically solved by Barnett [1.6] showing that the Abraham and Minkowski momenta are, respectively, the kinetic and canonical optical momenta. The answer to the Abraham-Minkowski debate does not decide which momentum is the true one, but rather to what physical reality each momentum component should be assigned.

[1.1] Bethune-Waddell, M. & Chau, K. J. Simulations of radiation pressure experiments narrow down the energy and momentum of light in matter. *Rep. Prog. Phys.* **78**, 122401 (2015).

[1.2] Mansuripur, M. Force, torque, linear momentum, and angular momentum in classical electrodynamics. *Appl. Phys. A* **123**, 653 (2017).

- [1.3] She, W., Yu, J. & Feng, R. Observation of a push force on the end face of a nanometer silica filament exerted by outgoing light. *Phys. Rev. Lett.* **101**, 243601 (2008).
- [1.4] Mansuripur, M. Comment on "Observation of a Push Force on the End Face of a Nanometer Silica Filament Exerted by Outgoing Light". *Phys. Rev. Lett.* **103**, 019301 (2009).
- [1.5] Brevik, I. Comment on "Observation of a Push Force on the End Face of a Nanometer Silica Filament Exerted by Outgoing Light". *Phys. Rev. Lett.* **103**, 219301 (2009).
- [1.6] Barnett, S. M. Resolution of the Abraham-Minkowski dilemma. *Phys. Rev. Lett.* **104**, 070401 (2010).

Action taken: To be precise in the use of the terminology, we replaced in the paper the term ‘force density’ with the term ‘time-averaged force density’ wherever optical cycle-averaging has been performed.

2. For the Abraham stress, is not DE equal to ED and BH equal to ED? This would give the same stress as for the Minkowski case (as it should).

In general, the tensor product $\vec{D}\vec{E}$ ($\vec{B}\vec{H}$) is not equal to $\vec{E}\vec{D}$ ($\vec{H}\vec{B}$), but under the assumption of the constitutive relation $\vec{D} = \epsilon_0 \epsilon_r(\vec{r})\vec{E}$ ($\vec{B} = \mu_0 \mu_r(\vec{r})\vec{H}$), $\vec{D}\vec{E}$ ($\vec{B}\vec{H}$) does become equal to $\vec{E}\vec{D}$ ($\vec{H}\vec{B}$).

In the case of our model of the mirror, both the AB and MN stress tensors become equal. In the literature, often the stress tensors for the AB and MN formulations already have an identical form, such as the one appearing in the first line in Table 1. See also the footnote on page 7 of [1.2].

Action taken: None.

3. It would be useful to discuss PRL 101 243601 (2008) that is related, but performed with much simpler means, and controversial. What does the present study imply?

We have added the She, Yu and Feng paper [1.3] to the main reference list of our paper. We have devoted some discussion to [1.3] already in response to the first comment of reviewer #1. With respect to [1.3], our present study implies that the interpretation of the mechanical motion of matter due to the transfer of optical momentum requires more involved descriptions of momentum transfer, such as the coupling of the electromagnetic theory and elasticity or fluid dynamics [1.7]. To access the small nuances between the formalisms, one has to access quantities sensitive to the force density distribution, such as elastic wave propagation as measured in our paper, and not just to the total force, such as the experiment presented in the She, Yu and Feng paper [1.3].

- [1.7] Leonhardt, U. Abraham and Minkowski momenta in the optically induced motion of fluids. *Phys. Rev. A* **90**, 033801 (2014).

Action taken: None.

4. Is the Einstein-Laub momentum not just the Abraham momentum with electrostriction/dipole force included? See PRA 90, 033801 (2014) for the case of fluids (solids should be similar). One might expect this to be the most promising candidate for explaining the momentum transfer.

The Einstein-Laub (EL) formalism leads to the following form of the time-averaged force density $\frac{1}{4}\epsilon_0(\epsilon_r(\vec{r})-1)\bar{\nabla}E(\vec{r})^2$ which differs from the AB and MN time-averaged force density $-\frac{1}{4}\epsilon_0E(\vec{r})^2\bar{\nabla}\epsilon_r(\vec{r})$.

Even by including the following time-averaged electrostriction term $\frac{1}{12}\epsilon_0\bar{\nabla}\left((\epsilon_r(\vec{r})-1)(\epsilon_r(\vec{r})+2)E(\vec{r})^2\right)$ to the AB and MN force density in the usual, empirical way [1.8], the EL time-averaged force density still differs from the AB and MN time-averaged force density that includes the additional electrostriction term.

The EL time-averaged force density is a polarization-independent bulk force density that is proportional to the gradient of intensity. The AB or MN time-averaged force densities are non-zero only at the interfaces and point perpendicular the interfaces. They are also polarization independent and give a null lateral component, thus they do not account for electrostriction (and magnetostriction). On the contrary, the EL, AMP and Chu formalism do account for electrostriction and magnetostriction. We agree with Mansuripur [1.2] that since the EL formalism does not involve any hidden entities, this formalism comes handy to calculate the total force exerted by light on matter. Note that even though the EL formalism fully conforms to the conservation of energy and momentum, it lacks the stress-energy-momentum tensor invariance as shown by Kemp and Sheppard [1.9] and is for that reason not the correct field-kinetic subsystem since it violates the physical constraint of relativistic invariance.

[1.8] Hallanger, A., Brevik, I., Haaland, S. & Sollie, R. Nonlinear deformations of liquid-liquid interfaces induced by electromagnetic radiation pressure. *Phys. R. E* **71**, 056601 (2005).

[1.9] Kemp, B. A. & Sheppard, C. J. Electromagnetic and material contributions to stress, energy, and momentum in metamaterials. *Advanced Electromagnetics* **6**, 11-19 (2017).

Action taken: We have added the Leonhardt paper [1.7] (PRA 90, 033801 (2014)) to the main reference list of our paper.

Authors' response to Reviewer #2 remarks:

Reviewer #2 remarks are typeset in italic, magenta font.

Authors' response follows in upright, black font.

In this manuscript, Pozar et al show modeling and experiments of electromagnetic propagation and the conversion of energy and momentum into elastic waves in a solid material. A clever experimental technique is presented and simulations and experiments are shown to agree well. The results seem plausible and are supported by the data, but it would be helpful if the authors could more clearly state the new results that are presented in this manuscript, with respect to previous works. In the abstract they state that they have "compelling evidence," but they should be more explicit (and quantitative) about what is observed and what is different in this experiment, making in more "compelling" than others. As an example, at the end of the first paragraph the authors state that, elastic waves induced by radiation pressure "have yet to be directly measured." Then the next paragraph begins with "Elastic waves driven by light momentum can be detected as transient normal displacements on the surface of an illuminated object [10–13]," which sounds like they have been measured (see references 12, which is their own work)... I think there are new results in this manuscript, but those should be more obvious.

We would first like to express our appreciation to the reviewer for his/her valuable comments and insightful suggestions, which have helped us to improve the paper.

We have made several modifications in the Introduction and Discussion sections. In the introduction, it is now clearly stated what has already been achieved both theoretically and experimentally in the area of light-pressure-driven elastic waves. The novelty of this paper with respect to the previous works is now given explicitly.

What is compelling here is the good match between the measured displacement-waveforms caused by the light-pressure-induced elastic waves and the ab-initio modeling of the whole detection chain (light pulse -> light-matter interaction -> wave propagation -> input into the sensor -> sensor transfer function -> sensor output) that needs no fitting. By comparing the calculated displacement waveforms with the detected ones, one can conclude that the measured waveforms were caused predominantly by the normal component of the optical forces (radiation pressure) and that absorption-driven waves play a secondary role, since the time evolution of the thermoelastic waves differs both in shape (opposite polarity of peaks) and amplitude.

The apparent inconsistency of our claims has been rewritten as to remove any ambiguity in possible interpretations of the claims of this work.

Action taken: Several modifications have been made in the Introduction and Discussion sections as suggested by the reviewer #2.

My further comments are below.

We shall address your comments hereafter.

-In the abstract, the authors state that the "details" of the momentum conversion from EM fields to elastic waves within a solid have been predicted by theory, but that it has not been validated by

experiments because of heating due to light absorption. There is then very little text within the manuscript describing heating effects on momentum transfer or optomechanical forces. I think this should be discussed within the manuscript to put such effects into context. Two works that discuss and compensate for thermal effects in optical force/momentum experiments are:

(a) Weld et al, Appl. Phys. Lett. 89, 164102 (2006)

(b) Ma et al, Appl. Phys. Lett. 106, 091107 (2015)

The authors should consider adding a short discuss with respect to this.

The heating effects due to laser pulse absorption in the HR coating were described and compared to the measurements and the normal/lateral optical forces in the very last paragraph in Results section and particularly in the caption of Fig. 5. Details of the modeling of the thermally-induced waves are given in Methods subsections *Wave propagation modelling (finite element method)* and *Waveform modelling (statistically-enhanced Green's function formalism)*. The conclusion that radiation pressure is the dominant effect in launching the measured elastic waves is based on these results.

The papers by Weld and Kapitulnik [2.1] and by Ma, Garrett and Munday [2.2] use a modulated laser beam to excite the microcantilever bending by comparing the vibrational response of the cantilever near the fundamental resonance frequency of the cantilever (350 Hz [2.1] and 17.600 Hz [2.2]). As for the similarities of their work with our present work, they can also distinguish between the radiation-pressure force and the thermoelastic loading based on the frequency response and bending direction. However, their experiments have no access to elastic waves induced by either mechanism.

[2.1] Weld, D. M. & Kapitulnik, A. Feedback control and characterization of a microcantilever using optical radiation pressure. *Appl. Phys. Lett.* **89**, 164102 (2006).

[2.2] Ma, D., Garrett, J. L. & Munday, J. N. Quantitative measurement of radiation pressure on a microcantilever in ambient environment. *Appl. Phys. Lett.* **106**, 091107 (2015).

Action taken: We have expanded the text within the manuscript as suggested. A short discussion was added to the main text to explain the similarities of the work presented in [2.1] and [2.2] and our current work. Both references were added to the reference list.

-The authors mention the Abraham and Minkowski formalisms a few times and discuss, briefly, the controversy resulting from these two descriptions. This is a very interesting point and the authors state that these measurements may shed light on this problem. They say "...it implies that it can be possible to empirically validate an electrodynamic formalism through sufficiently precise measurement of the elastic waveforms generated by optical shear forces." But, how precise would the measurements need to be? Can they do it? I'd like to see something quantitative here.

-A similar comment appears at the end of the paper: "This work also forms a basis for experimental resolution of the Abraham-Minkowski controversy." How? What are the different, measureable features? Can this currently be resolved? How far off are the sensitivities? If you are not there yet, what would be needed to get there? These answers should be included in the discussion.

Since reviewer #3 also felt additional explanation is needed in this direction, we described two possible improvement paths that could enable direct access to validate or disprove a chosen EM formalism. We have largely expanded the Discussion section to include this interesting topic as suggested by reviewers #2 and #3.

We have also set a larger distance to the catchy term ‘Abraham-Minkowski controversy’. As for the original Abraham-Minkowski debate, this problem has already been theoretically solved by Barnett [2.3] showing that the Abraham (AB) and Minkowski (MN) momenta are, respectively, the kinetic and canonical optical momenta. Since the time-averaged force density distribution predicted by the AB or MN formalism is the same (under the assumptions of our model it equals $-\frac{1}{4}\epsilon_0 E(\vec{r})^2 \vec{\nabla} \epsilon_r(\vec{r})$ [2.4]), our experiments cannot differentiate between the two. The difference between the AB and MN instantaneous force density distribution is $\frac{\partial}{\partial t}(\vec{g}_{MN} - \vec{g}_{AB})$, a term oscillating at optical frequencies [2.4] which our detection technique cannot access due to its slower (MHz) response time. Here, $\vec{g}_{MN} = \vec{D} \times \vec{B}$ is the Minkowski momentum density, $\vec{g}_{AB} = \epsilon_0 \mu_0 \vec{E} \times \vec{H}$ is the Abraham momentum density, \vec{E} is the electric field, \vec{H} is the magnetic field, \vec{D} is the electric displacement field, \vec{B} is the magnetic flux density, ϵ_0 is the free-space permittivity, μ_0 is the free space permeability, $\epsilon_r(\vec{r}) = n(\vec{r})^2$ is the relative permittivity that equals the square of the refractive index $n(\vec{r})$, \vec{r} is the position vector, $\vec{\nabla}$ is the gradient operator, and $\frac{\partial}{\partial t}$ is a partial time derivative. To experimentally detect the difference between the AB and MN formalisms, one has to look for measurable effects from the oscillating term $\frac{\partial}{\partial t}(\vec{g}_{MN} - \vec{g}_{AB})$. On the other hand, our detection technique does offer access to the time-averaged force density distribution. Therefore, by improving the approach presented in this paper, we can discriminate among the EM formulations that give different time-averaged force density distributions. These facts are conveyed in Figs. 3 and 5.

[2.3] Barnett, S. M. Resolution of the Abraham-Minkowski dilemma. *Phys. Rev. Lett.* **104**, 070401 (2010).

[2.4] Mansuripur, M. Force, torque, linear momentum, and angular momentum in classical electrodynamics. *Appl. Phys. A* **123**, 653 (2017).

Action taken: Our claim that the proposed experimental method can be used to distinguish among the electrodynamic models is now described in detail and elaborated on in the last paragraph of the Discussion section.

Authors' response to Reviewer #3 remarks:

Reviewer #3 remarks are typeset in italic, green font.

Authors' response follows in upright, black font.

The work reports on detection of elastic waves generated following the reflection of electromagnetic waves over a non-absorbing dielectric plate due to the transmission of momentum from the field photons into the object. The authors design and develop an original and high sensitive experiment to detect the momentum driven mechanical waves under the condition of mitigated thermal effects. A set a piezoelectric detector conveniently located on the front a rear side of the device are able to detect pm-amplitude mechanical waves. Authors provide convincing data about the reality of the waves. They also use ab-initio electrodynamic models based on different descriptions of the spatial distribution of normal and shear force densities to explain the observed effects. The models describe qualitatively well the observations. Furthermore, authors claim that the proposed experimental method can be used to validate the correct electrodynamic model, although they do not provide a clear evidence of that. The work contains detailed information about the experimental methods and the mathematical models used to describe the electrostatics of the interaction. The manuscript is well organized and written in good English. Considering the fundamental experiment described and the originality of the contribution, I recommend the manuscript for publication.

We thank the reviewer for the recommendation for publication.

Action taken: Our claim that the proposed experimental method can be used to validate the correct electrodynamic model is now described in detail and elaborated on in the last paragraph of the Discussion section.

REVIEWERS' COMMENTS:

Reviewer #1 (Remarks to the Author):

I am happy with the response and the revised manuscript and recommend acceptance.

Reviewer #2 (Remarks to the Author):

The authors have satisfactorily addressed my questions/comments, and I recommend publication.

Reviewer #3 (Remarks to the Author):

Authors have responded to the previous criticism.

I recommend the paper for publication.